# A Simple and Cost-Effective DNA Preparation Method Suitable for High-Throughput PCR Quantification of Hepatitis B Virus Genomes

**DOI:** 10.3390/v12090928

**Published:** 2020-08-24

**Authors:** Eunji Jo, Jaewon Yang, Alexander Koenig, Seung Kew Yoon, Marc P. Windisch

**Affiliations:** 1Applied Molecular Virology Laboratory, Institut Pasteur Korea, 696 Sampyeong-dong, Bundang-gu, Seongnam-si, Gyeonggi-do 13488, Korea; eunji.jo@ip-korea.org (E.J.); jaewon.yang@ip-korea.org (J.Y.); alexander.koenig@ip-korea.org (A.K.); 2Catholic University Liver Research Center, The Catholic University of Korea, Seoul 06591, Korea; yoonsk@catholic.ac.kr; 3Division of Gastroenterology and Hepatology, Department of Internal Medicine, College of Medicine, Seoul St. Mary’s Hospital, The Catholic University of Korea, Seoul 06591, Korea; 4Division of Bio-Medical Science and Technology, University of Science and Technology, 217, Gajeong-ro, Yuseong-gu, Daejeon 34113, Korea

**Keywords:** hepatitis B virus, viral DNA, genome equivalents, intracellular, extracellular, DNA preparation, qPCR, microtiter plates, antivirals

## Abstract

Hepatitis B virus (HBV) is a para-retrovirus that reverse transcribes its pregenomic RNA into relaxed circular DNA inside viral nucleocapsids. The number of HBV genomes produced in vitro is typically quantified using commercial silica-membrane-based nucleic acid purification kits to isolate total DNA followed by HBV-specific quantitative PCR (qPCR). However, despite the convenience of commercial kits, this procedure is costly and time-consuming due to multiple centrifugation steps, which produce unnecessary waste. Here, we report a rapid, cost-effective, and environmentally friendly total DNA preparation method. The assay is based on the simple incubation of detergent and proteinase K with cells or cell-free supernatants to permeabilize cells and disrupt viral particles. After heat inactivation and subsequent centrifugation to clear the lysates, DNA samples are directly subjected to qPCR to quantify HBV genomes. As a proof of concept, the assay was developed in 12-well plates to assess intra- and extracellular HBV genome equivalents (GEqs) of stably viral-replicating cell lines (e.g., HepAD38) and HBV-infected HepG2-NTCP cells, both treated with lamivudine (LMV), an HBV replication inhibitor. Viral DNA was also prepared from the serum of patients chronically infected with HBV. To validate the assay, a representative commercial DNA isolation kit was used side-by-side to isolate intra- and extracellular HBV DNA. Both methods yielded comparable amounts of HBV GEqs with comparable LMV 50% efficient concentration (EC_50_) values. The assay was subsequently adapted to 96- and 384-well microtiter plates using HepAD38 cells. The EC_50_ values were comparable to those obtained in 12-well plates. In addition, the calculated coefficient of variation, Z’ values, and assay window demonstrated high reproducibility and quality. We devised a novel, robust, reproducible, high-throughput microtiter plate DNA preparation method suitable for quantifying HBV GEqs by qPCR analysis. This strategy enables rapid and convenient quantitative analysis of multiple viral DNA samples in parallel to investigate intracellular HBV replication and the secretion of DNA-containing viral particles.

## 1. Introduction

Hepatitis B virus (HBV) is one of the most common viral infections affecting humans worldwide. Currently, 3.5% of the world’s population is chronically infected with the virus and at risk of developing life-threatening liver diseases [1]. Although effective vaccines and drugs are available, chronic hepatitis B (CHB) is incurable, necessitating the development of new treatment strategies.

HBV is the prototype member of the family *Hepadnaviridae*. This small, enveloped, DNA-containing virus replicates inside viral capsids via reverse transcription of its pregenomic RNA (pgRNA), mediated by viral reverse transcriptase (RT), into an approximately 3.2-kb, partially double-stranded, relaxed circular DNA (rcDNA) genome [2]. Together with viral antigens, rcDNA is one of the most commonly used biomarkers in hospitals and laboratories (e.g., to determine viral infection status, evaluate viral fitness and antiviral efficacy of drugs and inhibitors, and guide treatment decisions) [3]. In the case of treatment complications, sequencing of the HBV rcDNA can provide insights into the potential development of resistance mutations [4]. Furthermore, in epidemiologic studies, HBV rcDNA sequences are informative regarding the distribution of different HBV genotypes and subgenotypes.

Isolation and purification of HBV DNA is a standard procedure widely used in research and diagnostic laboratories around the world. Thus, the procedure requires standardization, such as the use of specialized viral DNA isolation kits to ensure appropriate and standardized quantification of HBV genome equivalents (GEqs) by quantitative real-time PCR (qPCR). The majority of commercial DNA isolation kits are based on the use of silica membranes that bind nucleic acids. Small, centrifugeable, plastic silica-containing columns are loaded with either patient serum, cell lysate, or cell culture supernatant and allowed to bind DNA, followed by multiple wash steps using various buffers and ethanol until DNA is eluted. Each wash step, the elution step, and the binding of DNA require labor-intensive and time-consuming pipetting and centrifugation steps that constantly occupy the experimenter. Despite the convenience of using premixed reagents and dedicated columns in the DNA isolation kits, the majority of research laboratories opt for more rapid and cost-effective solutions to isolate and analyze viral genomes. Particularly when the number of samples is greater than 24, the usefulness of centrifugation-based, commercial, low-throughput DNA purification kits declines due to limited centrifuge capacity. For high-throughput DNA extraction, automated robotic systems can be used to provide high-quality genomic DNA, with the advantages of less hands-on time, greater consistency, and reduced probability of manual errors. However, for most basic research laboratories in need of high- or medium-throughput HBV DNA preparation, the purchase of a liquid automation station exceeds project funding.

Here, we report the development of a column-free, simple, low-cost total DNA extraction method suitable for the preparation of HBV DNA at high purity, sufficient for accurate quantification by PCR and sequencing. Our goal was to reduce the number of centrifugation and pipetting steps and upscale the throughput to 96- and 384-well plates, which can be analyzed simultaneously. An additional goal was to reduce the environmental footprint of the assay by limiting the use of plastic tips and columns and decreasing the amount of liquid waste (flow-through) generated.

## 2. Materials and Methods

### 2.1. Cell Culture and Chemicals

HepAD38 cells were kindly provided by Dr. Christoph Seeger (Fox Chase Center; Philadelphia, PA, USA) and cultured in Dulbecco’s Modified Eagle Medium (DMEM)/F12 medium supplemented with 10% fetal bovine serum (FBS), 50 U/mL penicillin-streptomycin (P/S), and 400 µg/mL geneticin. HepG2-NTCP cells were generated and cultured in DMEM supplemented with 10% FBS, P/S, and L-glutamine, as described previously [5]. Lamivudine (LMV) was purchased from Selleckchem (Houston, TX, USA).

### 2.2. HBV Infection

Cells were infected with HBV, as previously described [5]. In brief, HepG2-NTCP cells were infected with 1000 GEqs/cell of HepAD38-derived HBV in the presence of 4% PEG8000 (Sigma-Aldrich, St. Louis, MO, USA). At 1 day post-infection (dpi), input HBV was washed out using 1× DPBS, and at 7 dpi, the cells or supernatants were harvested for DNA preparation.

### 2.3. Column-Free DNA Preparation

HepAD38 cells were seeded at densities of 1 × 10^6^, 1 × 10^5^, and 2.5 × 10^4^ cells/well in 12-, 96-, and 384-well plates, respectively. Next, 100, 30, and 10 µL of supernatant of total volume 2000, 100, and 50 µL from these plates, respectively, was used for lysis. Cells in 1× DPBS or cell-free supernatant (cell culture supernatants and patient serum) were incubated with 1 volume of 2× proteinase K (proK) lysis buffer (100 mM Tris-Cl (pH 8.5), 2 mM EDTA, 1% sodium dodecyl sulfate (SDS), 400 µg/mL proteinase K (Cat No. PB0451, Bio Basic Inc., Toronto, Canada)) at 56 °C for 3–6 h (3 h for supernatants, 6 h for cells), followed by inactivation at 95 °C for 10 min. Then, 1.5 to 2 µL of the sample was used for qPCR reaction. All centrifugation steps were conducted at room temperature.

### 2.4. Column-Based DNA Preparation

Commercially available kits were used according to the manufacturer’s instructions. A QIAamp DNA Mini kit (Qiagen, Hilden, Germany) was used for cells, whereas a QIAamp MinElute Virus Spin kit (Qiagen) was used for cell-free supernatants. The starting volume of samples was equal to that in the column-free method, but the final elution volume was different (1- to 4-fold of the starting volume, meaning a factor of 1 to 4 of dilutions). The dilution factors at the end of sample preparation were used for the normalization of PCR quantification.

### 2.5. Quantitative Real-Time PCR for DNA Quantification

HBV DNA was analyzed by qPCR as previously described using a ViiA 7 system (Life Technology, Carlsbad, CA, USA) [6]. In brief, Premix Ex Taq (Cat No. RR390A, Takara Bio Inc., Shiga, Japan) was used for qPCR reactions. Sequences of primers and probes were as follows: forward primer: 5’-ACTCACCAACCTCTTGTCCT-3’, reverse primer: 5’-GACAAACGGGCAACATACCT-3’, and probe: 5’-FAM-TATCGCTGGATGTGTCTGCGGCGT-TAMRA-3’ [7]. For qPCR using HBV DNA prepared using the column-free method, 5% Tween-20 was added to the reaction mixture.

### 2.6. Patient Serum

Four samples of crude serum from patients chronically infected with HBV genotype C in South Korea were provided by Seoul St. Mary Hospital (Seoul, South Korea). Patients’ consent was obtained to use their samples for research purposes. The titers of HBV measured in the hospital were as follows: high titer patient 1 (HTP1), 4.2 × 10^6^; HTP2, 6.6 × 10^6^; low titer patient 1 (LTP1), 3.7 × 10^5^; LTP2, 5.1 × 10^5^ IU/mL. Furthermore, 30 µL of 2-fold diluted patient serum was used for DNA purification using column and column-free methods.

## 3. Results

### 3.1. Development and Validation of Column-Free DNA Preparation for HBV Genome Quantification

To develop a simple and cost-effective HBV DNA preparation method suitable for quantifying viral DNA by standard qPCR analysis, we devised a “column-free” DNA preparation method. Compared to conventional DNA isolation methods using commercially available kits that are commonly based on silica column and involve lysis, binding, and multiple DNA washing, centrifugation, and elution steps (Figure 1A), the column-free method is simpler because the binding–washing–elution steps are excluded. Moreover, the column-free method requires only a short spin-down to clarify samples, thus reducing the dependence on centrifugation (Figure 1B). We developed and validated conditions to treat cells or cell-free supernatants using lysis buffer containing 1% SDS and 400 µg/µL proteinase K (proK) in reference to the conventional lysis condition for the viruses [8]. Cells or supernatants harvested from large multi-plates (12-well plates) were incubated with lysis buffer at 56 °C for 3–6 h. After inactivation of the reaction at 95 °C for 10 min, followed by a short centrifugation step to spin down small volumes of liquids that may have collected inside of the tubes, samples were ready for qPCR analysis. To neutralize the high SDS concentration in the lysis buffer, we added 5% Tween-20 to the qPCR reaction, which had no effect on the reaction (Appendix A).

Next, as a proof of concept, we performed a side-by-side comparison of the quantification of HBV DNA isolated by commercial columns and our column-free method. HepAD38 cells that stably replicate HBV were plated in 12-well plates applicable for low-throughput experimental layouts. We isolated intra- and extracellular HBV DNA using the column and column-free methods and determined the number of viral genomes by qPCR (Figure 2A). A total of 6.8 × 10^7^ and 9.8 × 10^5^ copies/µL of intra- and extracellular HBV DNA, respectively, were prepared using the column-free method. DNA amounts obtained by the column-free method were similar but slightly higher than the amounts of intra- and extracellular DNA prepared using columns (1.9 × 10^7^ and 5.9 × 10^4^ copies/µL of intra- and extracellular DNA, respectively). The qPCR samples derived from cell lysates prepared by either preparation method consisted of total cellular DNA, including different forms of HBV genomes, rcDNA, covalently closed circular DNA (cccDNA), and chromosome-integrated HBV DNA. The lower number of HBV DNA copies quantified using the column-based isolation method may have been due to a loss of DNA during washing and elution steps, considering that the yield of the silica column is not always 100% and depends on the sample volume and virus titer.

In addition, we compared the assay sensitivity window (signal-to-background ratio) using the HBV DNA replication inhibitor lamivudine (LMV). Similar assay windows (DMSO-to-LMV treated) were obtained by applying the two DNA preparation methods, indicated by 170- and 130-fold reductions in intracellular DNA and 70- and 85-fold reductions in secreted, extracellular DNA prepared using the column and column-free methods, respectively. The qPCR background of the column-free and column-based assays depends on the experimental settings. Using cell lysates of LMV-treated HepAD38 or infected HepG2-NTCP cells, HBV replication templates in the form of cccDNA, chromosomal integrated HBV DNA, as well as residual replication products of rcDNA molecules were detected as background. In contrast to HepAD38 cells, a higher background was detected for HBV-infected HepG2-NTCP cells, which is most likely due to the residual inoculum of viral rcDNA genomes that were not removed during wash steps. To evaluate whether the methods affect the 50% effective concentration (EC_50_) of the inhibitor, cells stably replicating HBV were treated with different concentrations of LMV. The resulting EC_50_ values were comparable over the range of 0.008–0.02 µM (Figure 2C, Table 1). After validating the column-free DNA preparation method using stably HBV-replicating cells, we further validated the assay using HBV-infected HepG2-NTCP cells, once again comparing the column-free and column-based methods side-by-side. As demonstrated above, a similar pattern of quantified HBV DNA (Figure 2B), as well as comparable LMV EC_50_ values of 0.011 ± 0.005 µM (Intra) and 0.016 ± 0.007 µM (Extra) on average were observed for the two DNA preparation methods, indicating that the column-free DNA preparation method is applicable to antiviral testing (Figure 2D).

Finally, we examined the feasibility of preparing HBV DNA from CHB patient-derived serum samples using the column-free method. Four patient serum samples with high and low viral titer were obtained, and viral DNA was prepared using the column-free method and a column-based kit as a reference. Both preparation methods yielded comparable amounts of HBV DNA, demonstrating that even high plasma protein concentrations are tolerated by the column-free proteinase K digestion protocol (Figure 2E).

### 3.2. Upscaling of the Column-Free DNA Preparation Method to Medium- and High-Throughput Quantification of Intra- and Extracellular HBV Genomes

After the demonstration of proof of concept using 12-well plates, we upscaled our novel DNA preparation method for medium- and high-throughput analyses by miniaturization of the assay for 96- and 384-well microtiter plates (Figure 3A). Thus, we plated HepAD38 cells in 96- or 384-well microtiter plates and treated the cells with DMSO or 10 µM LMV. After 1 week, the intra- and extracellular HBV DNA was isolated from the cells and supernatants using the above-described column-free method by incubation with proK lysis buffer in new microtiter plates. For quantification of genomic HBV DNA, 2 µL each of cleared lysates and supernatants was directly subjected to qPCR in suitable real-time PCR plates of the same format. We analyzed 30 and 90 replicative samples in 96- and 384-well plates, respectively, which are presented as single dots at their C_t_ values in Figure 3B and Figure 3C. To evaluate assay robustness, we calculated the percent coefficient of variation (%CV), Z’ value, and assay windows, as summarized in Table 2. The results demonstrated highly robust DNA preparation and qPCR analysis using the microtiter plates, as indicated by %CVs of 0.87–2.96, Z’ values of 0.53–0.88, and a 30- to 1260-fold assay window. In addition, the LMV EC_50_ values were 0.039 ± 0.002 and 0.033 ± 0.008 µM for intra- and extracellular DNA, respectively, in 96-well plates and 0.066 ± 0.016 and 0.035 ± 0.023 µM for intra- and extracellular DNA, respectively, in 384-well plates. Analyses of the dose-response curves indicated no significant differences in EC_50_ values between the 96- and 384-well plates, and the values were comparable to those obtained using 12-well plates (Figure 3D, Figure 3E and Table 1).

## 4. Discussion

The quantification of viral genomes is commonly practiced in laboratories to assess viral replication and the effect of antiviral agents. Here, we report the development of a simple, low-cost total DNA preparation method suitable for the detection of HBV DNA by qPCR in research laboratories. Compared to commercially available standard silica-column-based low-throughput kits or vacuum- and centrifuge-based medium-throughput kits, we improved the time and cost efficiency by excluding multiple washing and centrifugation steps. Consequently, we could reduce the material costs and the number of pipetting steps and increased the throughput of samples that can be analyzed simultaneously using 96- or 384-well microtiter plates. We could therefore prepare HBV DNA for up to 384 individual HBV cell culture samples and accurately quantify the intra- and extracellular viral genomes. The only other method that provides comparable sample throughput per unit of time requires cost-intensive automated robotics, which are unaffordable to the majority of research laboratories and are commonly used only in diagnostic and other highly specialized laboratories. Furthermore, compared to column-based DNA purification kits, our column-free DNA preparation method requires fewer pipette tips, columns, tubes, and less buffer per sample, thereby significantly improving the environmental footprint.

Importantly, despite the simplicity of our column-free method, the quality/purity of the DNA prepared using the method is suitable for standard sequence analysis, with a read length of approximately 500 base pairs and a very low background (Appendix A). Moreover, the column-free method is suitable for preparing HBV DNA from serum samples obtained from chronically infected patients, as shown by the data reproducibility observed with both DNA preparation methods. These results demonstrate that the preparation of partially double-stranded viral DNA genomes from viral particles by stripping off the envelope and digesting the capsid using a mix of detergent and proteinase K provides DNA of a sufficient level of purity for accurate qPCR analysis and sequencing. The qPCR protocol utilized in this study tolerates DNA at this level of purity, thus rendering laborious and expensive column-based purification procedures unnecessary. Even with the low PCR template abundance of HBV genomes recovered from a single well of a 384-well plate, the window of measurement, assay robustness, and reproducibility were high, as shown by the average %CV and Z’ values of 2.2 and 0.63, respectively. These characteristics make the method suitable for high-throughput DNA quantification in applications such as screening campaigns for novel inhibitors of viral replication and secretion of DNA-containing virions. As a proof of concept, we demonstrated that column-free DNA preparation from a single well of a 384-well plate is possible and that the method is suitable for antiviral drug testing, as the LMV EC_50_ values were comparable regardless of the plate format. In contrast, column-based DNA purification kits are not suitable for medium- or high-throughput approaches and have a minimum and maximum capacity for DNA yield. As such, samples with too low or too high DNA abundance cannot be fully recovered, and this limitation in sample recovery yield can lead to inaccurate PCR quantification results.

Here, we report the successful development of a novel, simple DNA preparation method suitable for preparing a multitude of samples in parallel. The method provides DNA of quality suitable for qPCR and sequence analyses. The new method does not require DNA binding or washing steps and requires only a very short centrifugation step to clear the sample after proteinase K treatment. We speculate that the column-free DNA preparation method could be applicable to other DNA viruses and most likely could be applied to RNA viruses with some modifications. In summary, we developed an HBV DNA isolation assay that enables rapid, simple, low-cost, and environmentally friendly quantitative analysis of a multitude of samples in parallel, and that is applicable to medium- to high-throughput screening.

## Figures and Tables

**Figure 1 viruses-12-00928-f001:**
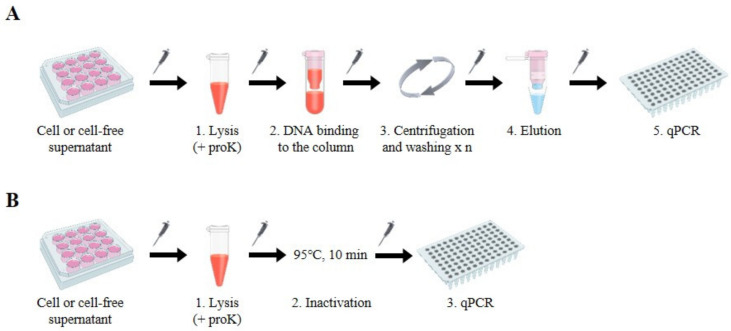
Schematic illustration of the HBV DNA preparation workflow for qPCR analysis. (**A**) Column-based DNA preparation method using a commercially available kit as follows: 1. Lysis in the presence of proteinase K (proK); 2. Binding of DNA to the column; 3. Centrifugation and addition of washing buffer (repeat step 3 two times); 4. Elution of DNA samples; 5. DNA quantification by qPCR. (**B**) Column-free DNA preparation method: 1. Cells or cell-free supernatants are incubated with lysis buffer containing proK at 56 °C for 3–6 h; 2. Inactivation at 95 °C for 10 min; 3. Quantification of DNA by qPCR.

**Figure 2 viruses-12-00928-f002:**
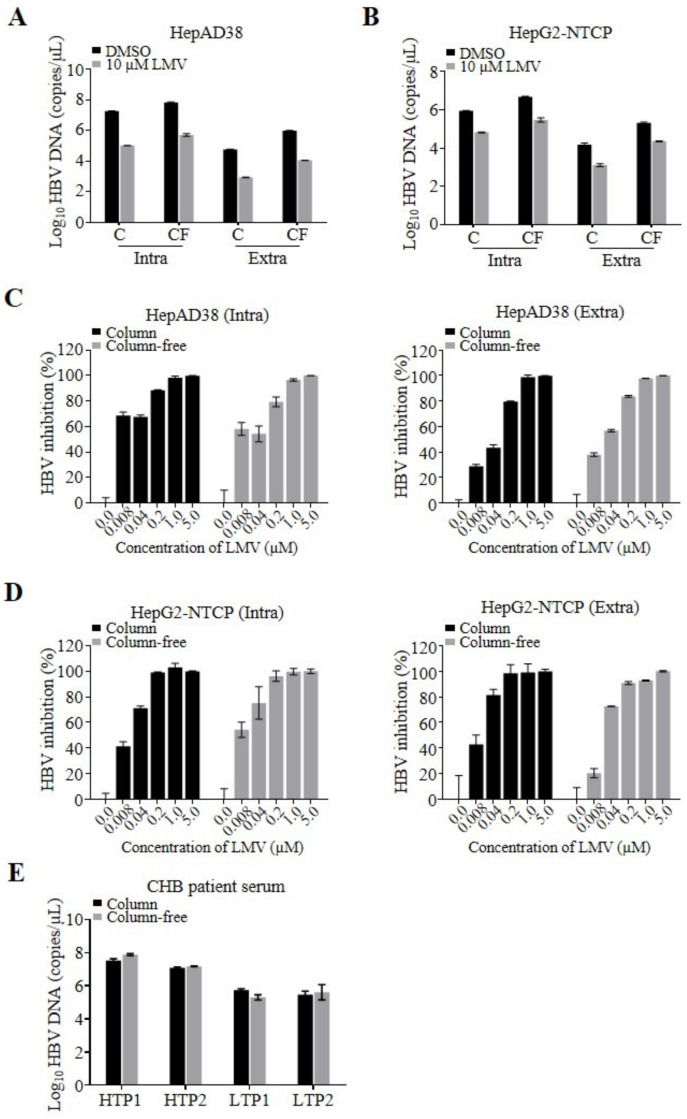
Assay development in 12-well plates. HepAD38 cells were seeded and treated with dimethyl sulfoxide (DMSO) or lamivudine (LMV) the following day. At 5 days post cell seeding, cells and supernatants were harvested for DNA preparation. HepG2-NTCP cells were infected with HBV at 1 day post cell seeding, and the next day viral inoculum was washed out, followed by DMSO or LMV treatment. At 7 days post-infection, cells and supernatants were harvested for DNA preparation. (**A**,**B**) qPCR analysis of HBV DNA isolated from HepAD38 (**A**) or HBV-infected HepG2-NTCP cells (**B**) and their culture supernatants in the presence of DMSO (black bars) or 10 µM LMV (gray bars) using the column (C) or column-free method (CF). (**C**,**D**) HepAD38 (**C**) or HBV-infected HepG2-NTCP cells (**D**) were treated with LMV at the indicated concentrations. HBV DNA from the cells (Intra, left) and supernatants (Extra, right) were prepared using the column (black bars) or column-free method (gray bars) and analyzed by qPCR. HBV inhibition (%) was normalized by DMSO set as 0% inhibition, and 10 µM LMV was set as 100% inhibition. Representative data of two independent experiments are shown as mean ± standard deviation (s.d.). (**E**) HBV DNA was extracted and isolated from the serum of patients chronically infected with HBV with high titer (HTP) or low titer (LTP) of virus using the column (black bars) or column-free (gray bars) method and analyzed by qPCR.

**Figure 3 viruses-12-00928-f003:**
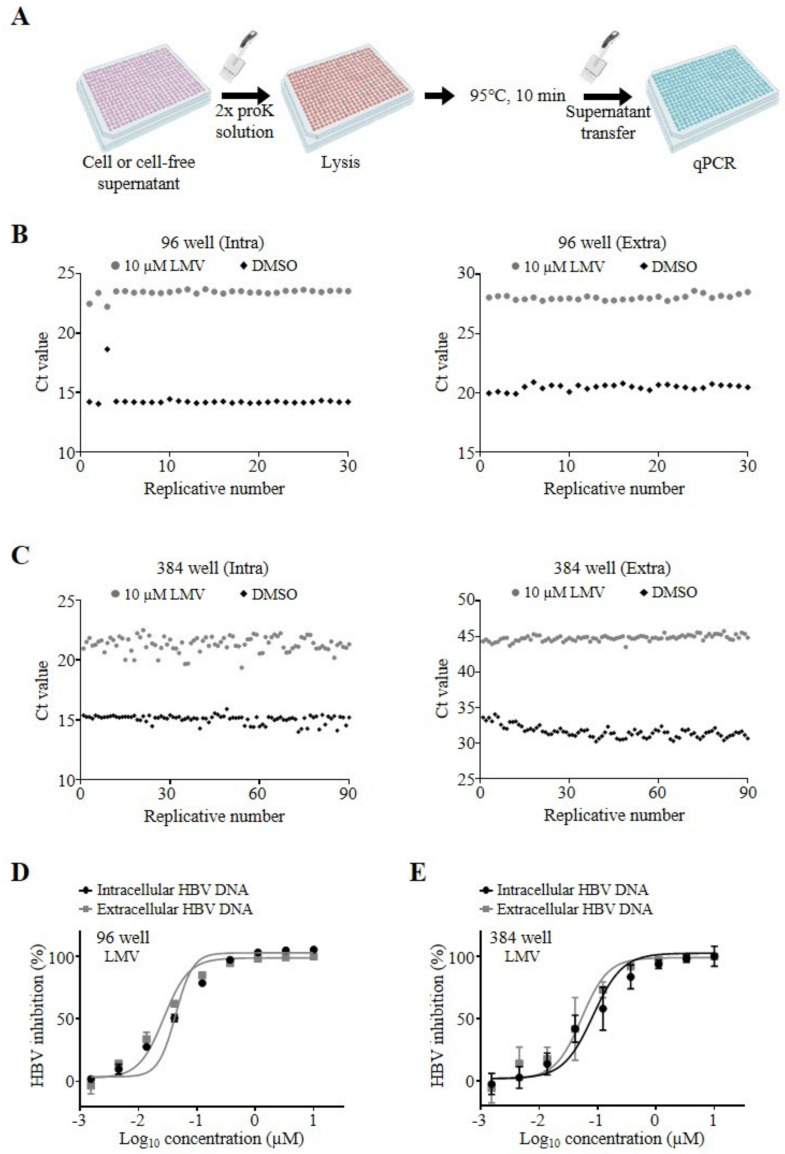
Assay validation in microtiter plates. (**A**) Schematic workflow of column-free DNA preparation in microtiter plates. HepAD38 cells were prepared and treated with LMV, as described above. Cells or cell-free supernatants in 96- or 384-well microtiter plates were lysed and subjected to the same plate format for qPCR analysis. HepAD38 cells were plated in 96- or 384-well microtiter plates at 1 × 10^5^ cells/well (**B**) and 2.5 × 10^4^ cells/well (**C**), respectively. Hepatitis B virus (HBV) DNA was prepared from cells (Intra, left) or supernatants (Extra, right) using the column-free method and analyzed by qPCR. Replicative numbers and C_t_ values of controls treated with DMSO (black diamonds) or 10 µM LMV (gray circles) are depicted as scatter plots on the *x*- and *y*-axes, respectively. (**D**,**E**) Dose–response curve analysis of intra- (black circles) and extracellular (gray squares) HBV DNA by LMV treatment of HepAD38 cells in 96- (**D**) or 384-well (**E**) plates. HBV DNA was analyzed by qPCR and normalized as described above. The concentrations of compounds in log scale and percent HBV inhibition are depicted on the *x*- and *y*-axes, respectively. Representative data of two independent experiments are shown as mean ± standard deviation (s.d.).

**Table 1 viruses-12-00928-t001:** Comparison of lamivudine 50% effective concentration (EC_50_) values.

Method	HBV DNA	12-Well	96-Well	384-Well
Column	Intra	0.008 ± 0.003	n.t. ^1^	n.a. ^2^
	Extra	0.014 ± 0.003	n.t. ^1^	n.a. ^2^
Column-free	Intra	0.018 ± 0.017	0.039 ± 0.002	0.066 ± 0.016
	Extra	0.020 ± 0.016	0.033 ± 0.008	0.035 ± 0.023

EC_50_ presented as mean ± standard deviation (µM).^1^ n.t. (not tested).^2^ n.a. (not available).

**Table 2 viruses-12-00928-t002:** Evaluation of assay performance in microtiter plates.

Plate Format	Sample	C_t_ Value		HBV Quantification (Copies/mL)		%CV ^1^		Z’ ^2^	Window ^3^
		DMSO	LMV	DMSO	LMV	DMSO	LMV		
96	Intra	14.20 ± 0.08	23.41 ± 0.30	9.47 × 10^10^ ± 3.95 × 10^9^	4.10 × 10^8^ ± 9.76 × 10^7^	2.56	0.91	0.88	230
	Extra	20.54 ± 0.17	28.06 ± 0.21	1.31 × 10^7^ ± 1.77 × 10^6^	4.71 × 10^4^ ± 6.80 × 10^3^	0.87	0.76	0.85	278
384	Intra	15.07 ± 0.35	21.33 ± 0.63	2.31 × 10^10^ ± 4.92 × 10^9^	7.58 × 10^8^ ± 1.88 × 10^8^	2.34	2.96	0.53	30
	Extra	31.60 ± 0.80	44.80 ± 0.41	3.53 × 10^6^ ± 1.33 × 10^5^	2.80 × 10^3^ ± 6.50 × 10^2^	2.56	0.92	0.72	1260

^1^ %CV (coefficient of variation). ^2^ Z’ determined as described by Zhang et al., 1999 [9]. 0.5 ≤ Z’ < 1 indicates “high quality” of the assay. ^3^ Window (signal-to-background ratio).

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
