# Peer review of "A Simple and Cost-Effective DNA Preparation Method Suitable for High-Throughput PCR Quantification of Hepatitis B Virus Genomes"

_viruses, 2020, doi:10.3390/v12090928_

Round 1
Reviewer 1 Report
HBV DNA quantification is a standard assay needed for HBV analyses, but purifying the DNA, particularly in 96- or 384-well formats can be laborious and expensive. Here, Jo et al report a simple technique for purifying cellular DNAs in multi-well plates that is suitable for qPCR analyses.
Strengths of this paper are its utility and simplicity. This approach will be widely applicable in many labs.
The small weaknesses include:
1) We developed a very similar assay a number of years ago that has become a mainstay of our lab’s operations. During development we found that different lots and vendors for the enzymes employed, particularly the qPCR mixes and protease, led to widely different success rates during DNA analysis. Most qPCR mixes could not tolerate the high contaminant level in the crude preps, and some lots of proteinase K could not be completely inhibited by the conditions employed by Jo et al. The authors should add a cautionary note about these complexities and provide the exact catalog number for the protease and qPCR mixes they used.
2) Describe the centrifugation step more clearly. How long, what G force, what temperature? Did the nuclei pellet out? Was the supernatant removed directly to the qPCR plate?
3) This assay does not appear remove the chromosomal DNA, which for cell lines like HepAD38 will yield a high background in qPCR that is equivalent to the cell number in the plate at the time of harvest. If cell numbers at harvest are similar to those at plating, the 96-well assay would have a minimal background of 1x10^5 copies per whatever final volume was recovered. Defining this background in the assay is needed. Note that it is quite possible that this background becomes irrelevant if the amount of lysate used for PCR is small and the viral titer is high, but that is not apparent in the document as it stands.
Minor issues:
- a) Please label the panels in Figs. 2 and 3 better so that they are interpretable without having to extract the data type from the legend. 2C and D need the units indicated in the X axis.
- b) Line 147: Confusing phrasing. I suggest changing “These numbers” to “DNA amounts obtained by the column method were similar but slightly lower….”
Reviewer 2 Report
In this technical note, Jo et al. present a simple, cost-effective and environmentally friendly DNA preparation method suitable for PCR quantification of total intracellular and extracellular HBV DNA. This is an interesting method that will be of interest for people working in the field. The manuscript is well written and easily comprehensible. I have few minor comments:
- In figure 1, the scheme of the experiment looks like the columns protocol is longer than the column-free one. However, this is not true since in the latest, an incubation of 3h-6h is performed to ensure lysis. So even though, they are more steps and require the actual presence of an experimenter, the columns protocol is shorter in time than the columns free one. So, I think that “rapid” line 78 or 120 is not so appropriate.
- Line 263, 264: authors showed similar results between columns and column-free protocols. They did not show a better DNA recovery nor increase sensitivity to improves the detection limit. This is an overstatement. Experiments with dilutions of samples should be performed to be able to keep this statement. This might actually be the opposite. Low amount of DNA might be easier to amplify when purity of the sample if better.
- In figure 2, it should be stated in the legend how many times with how many replicates the experiments have been performed. It should also be stated at what time post-seeding (for HepAD38) or post-infection (for HepG2-NTCP), cells have been treated with Lamivudine and for how long (same for Figure 3).
- Did the authors try to quantify cccDNA after the columns free protocol? Is it compatible with DNAse plasmide-safe, T5 or Exo I/III digestions? If so, it would make it even more attractive.
Reviewer 3 Report
In this manuscript, Jo et al. describe a method for the use of crude cell lysates in determining hepatitis B virus DNA load by qPCR. The manuscript is straightforward and well-written and describes a method that could significantly decrease costs associated with HBV research. While the use of crude lysates in qPCR quantitation is not a new development, its application to HBV research could prove useful. A major flaw in the current manuscript is inadequate description of protocols. Since this manuscript is focused mainly on a rather simple technique, the protocol should be extremely thorough, including all solution volumes, centrifugation times and speeds, incubation times and temperatures, cell numbers, well volumes, etc. The authors should edit so that the results can easily be replicated by other groups, and the assay can be used in other labs. The procedure for the column-based method does not need to be explicitly stated since it is a commercial kit with published instructions. Additional comments are below:
- One of the reasons the authors claim the assay offers more throughput than column-based DNA purification is less centrifugation. In section 3.1, the authors state there is a short centrifugation step after heat inactivation; however, the speed and time are not mentioned anywhere in the manuscript, and the centrifugation step is omitted from Figure 1. Please make sure all protocol steps are represented in the schematic.
- Line 147: “These numbers were similar but slightly lower compared to the amount of intra- and extracellular DNA prepared using columns…” This line is incorrect. The numbers were higher compared to columns.
- In Table 1, what is the difference between “not available” and “not tested?”
- Table 1: The 12-well LMV EC50 values for column-free have SD that is as large as the mean. Furthermore, Figure 2 shows that the concentrations of LMV did not go low enough to observe a proper dose response curve. For example, 8 nM LMV in figure 2C inhibited >60% of intracellular DNA production. Therefore, the reported EC50 values are not correct and should not be shown. Are the large SD in table 1 because of poor assay quality or because curves were fitted without proper drug concentrations? If EC50s are to be calculated, the experiments must be repeated with lower concentrations. The values for 96- and 384-well plates are much cleaner, likely because more concentrations were used, although the concentrations still did not go low enough to catch the bottom of the sigmoidal curve (figure 3D,E).
- Line 191: “After 1 week, the intra- and extracellular HBV DNA was isolated from the cells and supernatants using the above-described column-free method by incubation with proK lysis buffer in intermediate microtiter plates.” The method was never described in enough detail to surmise what was done here. Furthermore, the use of plates instead of tubes necessitates protocol changes, so please state the exact protocol used for 12-well plates and the exact protocol used for 96- and 384-well plates, including cell numbers, liquid transfers, etc.
- Figure 3A schematic is incomplete. Please fill in the schematic with all pertinent steps.
- The lysis procedure is by no means a novel development, so relevant publications should be cited that were used as inspiration.
Reviewer 4 Report
In this technical note, Jo E. et al., describe a simplified method for HBV DNA purification, very useful for high-throughput anti-viral drug screening and PCR quantitation of viral genome. The manuscript is very clear and well written.
Minor comments:
- please use the complete name of the person that gave you the HepAD38 cells (lane 86)
- provide the amount of starting material (the volume of supernatant and the number of cells) used for DNA purification in all cases, HepAD38 cells, HepG2-NTCP infected cells (in all plates format), or patients’ serum. I suppose that for column-free and column methods were used the same amount of starting material but specify this, please.
- lane 147, slightly higher instead of slightly lower.
